✿ PLOS | ONE

# Adaptation of H3N2 canine influenza virus to feline cell culture

**Haruhiko Kamiki, Hiromichi Matsugo, Hiroho Ishida, Tomoya Kobayashi-Kitamura, Wataru Sekine, Akiko Takenaka-Uema, Shin Murakami, Taisuke Horimoto** [ORCID] *

Department of Veterinary Microbiology, Graduate School of Agricultural and Life Sciences, The University of Tokyo, Bunkyo-ku, Tokyo, Japan

* ahorimo@mail.ecc.u-tokyo.ac.jp

## Abstract

H3N2 canine influenza viruses are prevalent in Asian and North American countries. During circulation of the viruses in dogs, these viruses are occasionally transmitted to cats. If this canine virus causes an epidemic in cats too, sporadic infections may occur in humans because of the close contact between these companion animals and humans, possibly triggering an emergence of mutant viruses with a pandemic potential. In this study, we aimed to gain an insight into the mutations responsible for inter-species transmission of H3N2 virus from dogs to cats. We found that feline CRFK cell-adapted viruses acquired several mutations in multiple genome segments. Among them, HA1-K299R, HA2-T107I, NA-L35R, and M2-W41C mutations individually increased virus growth in CRFK cells. With a combination of these mutations, virus growth further increased not only in CRFK cells but also in other feline fcwf-4 cells. Both HA1-K299R and HA2-T107I mutations increased thermal resistance of the viruses. In addition, HA2-T107I increased the pH requirement for membrane fusion. These findings suggest that the mutations, especially the two HA mutations, identified in this study, might be responsible for adaptation of H3N2 canine influenza viruses in cats.

## Introduction

Influenza A viruses are maintained in wild aquatic birds as natural reservoirs [1]. The waterfowl viruses are occasionally transmitted to poultry, causing several diseases. Although these avian viruses are rarely transmitted to mammals, specific subtypes of viruses have become endemic to humans and other mammals, including pigs and horses [2]. Among mammals, dogs and cats were thought to be historically restricted to influenza virus infections [3], although evidence of the infections in these companion animals have been sporadically reported [4–9]. Recently, two subtypes, H3N8 and H3N2, of canine influenza viruses (CIVs) have emerged and are endemic in dogs. H3N8 CIV was first detected among racing greyhounds in Florida, USA, in 2004, and studies revealed that the virus arose as a result of direct transmission of an H3N8 equine virus from a neighboring horse [10]. H3N2 CIV was first isolated in South Korea in 2007 and was revealed to be of avian virus origin [11]. Further studies showed that H3N2 CIV had been already present in southern China and South Korea before

**Data Availability Statement:** All relevant data are within the manuscript.

**Funding:** TH is supported by Grants-in-Aid for Scientific Research (A) (grant numbers:

18H03971) from the Japan Society for the Promotion of Science. The funder had no role in study design, data collection and analysis, decision to publish, or preparation of the manuscript.

**Competing interests:** The authors have declared that no competing interests exist.

2007 [12,13]. In 2012, the H3N2 CIV was isolated in Thailand [14], indicating that the virus had spread among the canine population in Asian countries. In 2015, H3N2 CIV caused an outbreak in an animal shelter in Chicago, USA, probably by importation of infected dogs from Asia [15]. Since then, H3N2 CIV infection has expanded nationwide in the USA, as has the H3N8 CIV [16]. Multiple subtypes of avian-origin viruses, such as H5N1, H5N2, H6N1, and H9N2, were isolated from dogs with respiratory illness in Asia since 2014 [17–19]. From 2013 to 2015, H1N1 swine origin-reassortant viruses were transmitted to dogs from pigs, and H1N1, H1N2, and H3N2 reassortant viruses of swine and canine H3N2 viruses were further detected in dogs in southern China [20].

Reports have shown that several human and avian influenza viruses can infect domestic cats. Serological surveys suggested that H3N2 human seasonal viruses can be transmitted to cats [21,22] and H1N1pdm2009 human virus managed to infect a cat colony, causing severe clinical diseases with fatalities [23]. The highly pathogenic H5N1 avian influenza virus (HPAIV) naturally infects domestic cats and causes severe illnesses [24]. In 2016, a low pathogenic strain of H7N2 AIV was transmitted into cats in an animal shelter and a veterinarian, who took care of these cats, got infected with the virus transmitted from the infected cats, causing mild respiratory symptoms [25]. In addition, under experimental conditions, domestic cats have been known to be susceptible to other influenza viruses such as H2N2 human, H7N7 seal, and H7N3 avian viruses [3,26,27].

Recently, transmission of H3N2 CIV into cats was also reported in two animal shelters in South Korea, where dogs and cats were accommodated together, resulting in 21.7% and 40% mortality of the cats in each shelter, respectively; the affected cats exhibited severe respiratory signs [28,29]. The transmission of H3N2 CIV into cats was also reported in China [20].

Mutation and reassortment are critical for cross-species transmission of influenza A virus, thereby establishing a new lineage in the new host. For example, H3N8 CIV emerged due to mutations in H3N8 equine virus [10], and H1N1pdm2009 virus emerged due to the reassortment between avian, swine, and human viruses with human-adaptive mutations [30]. Because of a frequent close contact between companion animals and humans, influenza viruses of companion animals pose a threat of being transformed into a pandemic virus by causing mutations or reassortment with human seasonal viruses. Indeed, a reassortant between H3N2 CIV and H1N1pdm virus emerged in a dog [31]. If a CIV adapts to cats and spreads in a cat population, the viruses can have more opportunities to be transmitted to humans, as they infect not only cats but also dogs. However, the molecular basis of H3N2 CIV adaptation to cats has not been elucidated.

In this study, to gain an insight into the mutations responsible for inter-species transmission of H3N2 CIVs from dogs to cats, we generated feline cell-adapted CIVs and analyzed the acquired mutations relevant to the viral adaptation.

## Materials and methods

### Cells and viruses

Madin–Darby canine kidney (MDCK) cells obtained from the American Type Culture Collection (ATCC; CCL-34) were maintained in minimal-essential medium (MEM) supplemented with 5% newborn calf serum (NCS) and antibiotics. Human embryonic kidney 293T cells from RIKEN BioResource Research Center (RCB2202) and fcwf-4 (*Felis catus* whole fetus) cells from ATCC (CRL-2787) were maintained in Dulbecco's modified Eagle's medium (DMEM) supplemented with 10% fetal bovine serum (FBS). Crandell–Rees feline kidney (CRFK) cells from ATCC (CCL-94) were maintained in DMEM supplemented with 5% FBS. All the cells were incubated in 5% $CO_2$ at 37˚C. A/canine/Madison/5/2015 (H3N2) virus,

kindly provided by Dr. Kathy Toohey-Kurth (Veterinary Diagnostic Laboratory, School of Veterinary Medicine, University of Wisconsin), was propagated in MDCK cells, and the supernatants containing viruses were collected and stored at -80˚C.

## Plaque assay

Confluent monolayers of MDCK cells were washed with MEM supplemented with 0.3% bovine serum albumin (MEM/BSA), infected with diluted virus, and incubated for 60 min at 37˚C. After the virus inoculum was removed, the cells were washed and overlaid with MEM/BSA containing 1% agarose supplemented with 1 μg/mL tosylsulfonyl phenylalanyl chloromethyl ketone (TPCK)-trypsin (Worthington, Lakewood, NJ, USA). The plates were incubated at 37˚C for 36 h, and then the cells were stained with crystal violet solution (0.1% crystal violet in 20% methanol).

## Adaptation of H3N2 canine influenza virus to CRFK cells

CRFK cells, seeded on three dishes of 3.5 cm diameter, were infected with 100 μL of 1000-fold-diluted stock virus ($3.5 \times 10^7$ PFU/mL) and maintained in DMEM supplemented with 0.3% BSA (DMEM/BSA) and 0.3 μg/mL TPCK-trypsin. At 48 h post-infection, the media containing viruses were centrifuged to remove cell debris and supernatants were collected. Next, three new dishes with CRFK cells were infected with viruses in 100 μL of 100-fold-diluted supernatants. Viruses obtained after 10 passages were collected as CRFK-adapted (CA) viruses for subsequent experiments.

## Sequence analysis

The RNAs of wild-type and CA viruses were extracted from the supernatants using ISO-GEN-LS (Nippon Gene, Tokyo, Japan). Reverse transcription of viral RNA was performed using primers containing the conserved sequences at the 3-prime ends of the viral segments using ReverTra Ace (TOYOBO, Osaka, Japan). Later, polymerase chain reaction (PCR) was conducted using specific primer pairs for each gene segment [32]. PCR products were purified using the Fast Gene Gel/PCR Extraction Kit (NIPPON Genetics, Tokyo, Japan) and sequenced directly using specific primers in an automated sequencer (Life Technologies Japan, Applied Biosystems 3130xl, Tokyo, Japan).

## Reverse genetics

The cDNAs of eight viral segments obtained from A/canine/Madison/5/2015 or CA viruses were cloned into the genomic RNA expression plasmid, pHH21 [33]. Escherichia coli DH5α cells transformed with the cDNA-inserted plasmids were incubated at 37˚C on LB agar plates supplemented with ampicillin. All the plasmids were sequenced to ensure that unexpected mutations were not present. We also generated viral polymerase and NP expression plasmids by inserting the coding sequences of A/WSN/33 segments into pCAGGS. Wild-type and mutant viruses were generated by reverse genetics as described previously [33]. Briefly, viral RNA expression, viral protein PB2-, PB1-, PA-, and NP-expression plasmids were mixed with a transfection reagent (TransIT-293; Mirus Bio, Madison, WI, USA), incubated at 21˚C for 15 min, and the mixtures were added to 293T cells. Transfected cells were incubated in Opti-MEM (Life Technologies/GIBCO, Grand Island, NY, USA) for 48 h. Supernatants containing the infectious viruses were harvested and propagated in MDCK cells maintained in MEM/BSA supplemented with 1 μg/mL TPCK-trypsin for 48 h, and then the virus-containing supernatants were aliquoted and stored at -80˚C.

## Virus growth kinetics in cultured cells

MDCK cells were infected with viruses at a multiplicity of infection (moi) of 0.001 and maintained in MEM/BSA with 1 μg/mL TPCK-trypsin. CRFK and fcwf-4 cells were infected with viruses at an moi of 0.01 and maintained in DMEM/BSA with 0.3 μg/mL and 0.4 μg/mL TPCK-trypsin, respectively. Supernatants were collected every 12 h and virus titers were measured using plaque assay.

## Fusion assay

Cell fusion assay was performed as previously described [34]. The cDNA of HA of wild-type virus or mutant viruses bearing HA1-K299R or HA2-T107I mutations were cloned into the protein expression plasmid, pCAGGS. All the plasmids were sequenced to ensure that unintended mutations were not present. MDCK cells were co-transfected with wild-type HA or mutant HA (HA1-K299R or HA2-T107I) expression plasmids, as well as with a fluorescent protein (Venus) expression plasmid (pcDNA3.1-Venus) by using polyethylenimine (PEI; Polysciences, Inc., Warrington, PA, USA). After transfection, the cells were incubated at 37°C for 24 h. The cells were then washed twice with $Mg^{2+}$ and $Ca^{2+}$ containing phosphate-buffered saline (PBS+) and treated with 5 μg/mL of TPCK-trypsin for 5 min at 37°C. Later, the trypsin was inactivated by washing with 4% NCS in PBS+. To induce cell fusion, the cells were treated with pH-adjusted PBS+ by citric acid for 1 min and incubated in growth medium at 37°C for 60 min. The fused cells were observed under a fluorescence microscope (Axio Vert.A1: ZEISS, Oberkochen, Germany).

## Thermostability assay

Thermostability assay was conducted as previously described [35]. Briefly, wild-type virus or mutant viruses bearing HA1-K299R or HA2-T107I were diluted to $4 \times 10^6$ PFU/mL and divided into six aliquots of 180 μL each. All the aliquots were heated at 42°C for 0, 15, 30, 60, 90, and 120 min, and then placed on ice immediately. Infectivities of the heated viruses were determined by plaque assay.

## Biosafety statement

All experiments with infectious viruses were approved by the University of Tokyo and conducted in BSL2 laboratories.

# Results

## Adaptation of an H3N2 CIV to CRFK cells

We generated feline cell-adapted H3N2 CIVs. A CIV strain was serially passaged 10 times in CRFK cells at 2-day intervals. After the 4th passage of these viruses, cytopathic effects were clearly visible, and rapidly expanded in the whole area of cells, leading to detachment of infected cells from the surface of the plate at the 9th passage. Subsequently, we used three lines of viruses from the 10th passage onwards as CRFK-adapted CIVs for further analyses, referred as Mad-CA1, -CA2, and -CA3. We compared the growth kinetics of wild-type and Mad-CA viruses in CRFK cells (Fig 1). Mad-CA viruses showed significantly higher titers than that of wild-type virus at each time point, indicating that these viruses were adapted to feline cells.

Interestingly, we were unable to obtain any CIVs that adapted to human-derived A549 or monkey-derived Vero cells (data not shown).

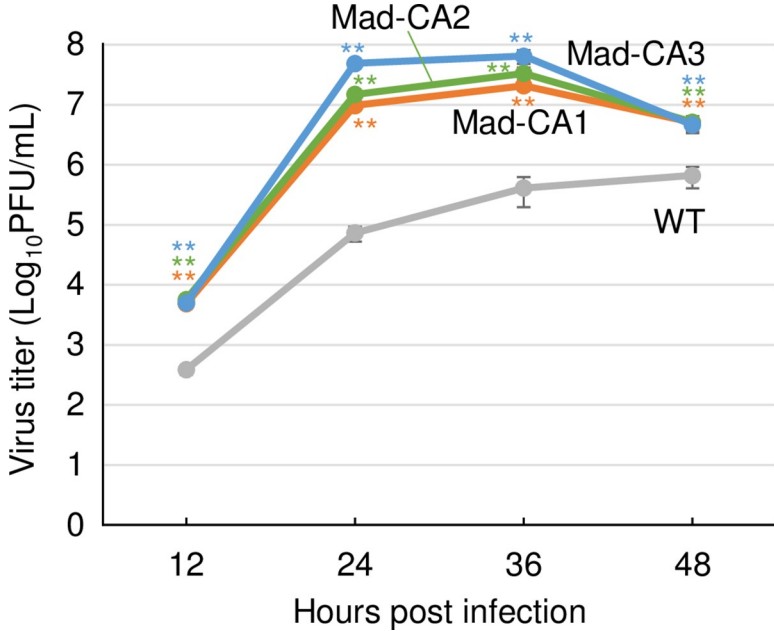

**Fig 1. Growth kinetics of CRFK cell-adapted viruses in CRFK cells.** CRFK cells were infected with CRFK cell-adapted viruses (Mad-CA1, -CA2, or -CA3) or wild-type (WT) virus at an moi of 0.01, supernatants were collected every 12 h, and virus titers of supernatants were measured. Error bars represent standard deviation (SD) of the means from three independent experiments. The statistical differences in the growth between the CRFK cell-adapted and wild-type viruses were assessed by two-tailed Student's t-test ($**p < 0.01$).

## Sequencing of CRFK cell-adapted CIVs

To identify the amino acid substitutions in the Mad-CA viruses, their complete genomes were sequenced and compared to those of the wild-type virus (Table 1). There were several nonsynonymous mutations in HA, NP, NA, and M segments. Notably, no mutation was found in the other segments including polymerase genes, which typically undergo several mutations during interspecies transmission of the virus [36]. Among them, we focused on mutations of envelope proteins such as HA1-K299R, HA2-T107I, NA-L35R, and M2-W41C mutations, three of which were observed in two or more Mad-CA viruses.

## Mutations responsible for CIV adaptation to CRFK cells

To characterize the mutations observed in the Mad-CA viruses, we generated viruses possessing single mutation (HA1-K299R, HA2-T107I, NA-L35R, or M2-W41C), viruses possessing double mutations (HA1-K299R/NA-L35R, or HA1-K299R/M2-W41C), and a virus possessing triple mutations (HA1-K299R/NA-L35R/M2-W41C), in addition to the wild-type virus by

**Table 1. Amino acids at the positions where mutations were observed in CRFK-adapted CIVs.**

| Virus | Segment | | | | | | |
|---|---|---|---|---|---|---|---|
| | HA | | NP | | NA | M | |
| | HA1-299 | HA2-107 | 289 | 309 | 35 | M1-15 | M2-41 |
| Wild-type | K | T | Y | N | L | I | W |
| Mad-CA1 | K | I | C | N | R | I | C |
| Mad-CA2 | R | T | Y | N | R | T | C |
| Mad-CA3 | R | T | Y | S | R | I | C |

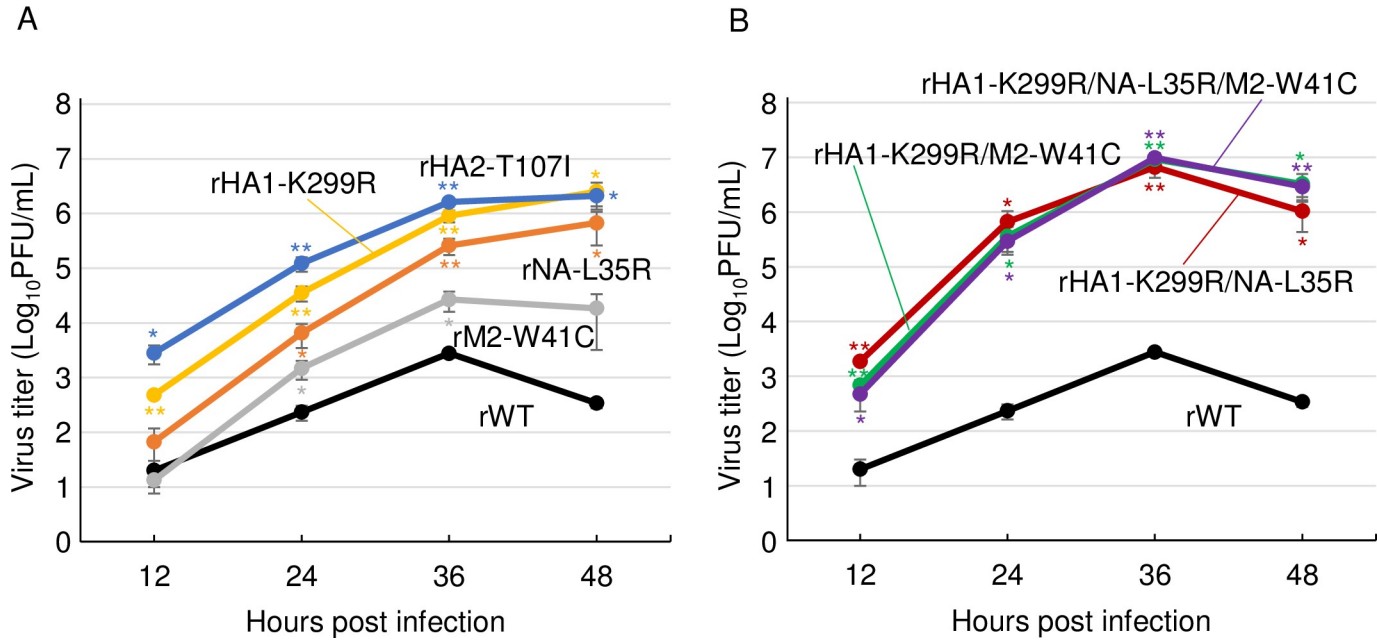

**Fig 2. Growth kinetics of the mutants generated by reverse genetics in CRFK cells.** CRFK cells were infected with recombinant viruses possessing (A) single mutation (rHA1-K299R, rHA2-T107I, rNA-L35R, or rM2-W41C), (B) multiple mutations (rHA1-K299R/NA-L35R, rHA1-K299R/M2-W41C, or rHA1-K299R/NA-L35R/M2-W41C), or recombinant wild-type (rWT) virus at an moi of 0.01. The virus titers were measured at each time point. Error bars represent SD of the means from three independent experiments. The statistical differences in the growth between the mutant viruses and rWT were assessed by two-tailed Student's t-test ($^*p<0.05$, $^{**}p<0.01$).

reverse genetics, referred as rHA1-K299R, rHA2-T107I, rNA-L35R, rM2-W41C, rHA1-K299R/NA-L35R, rHA1-K299R/M2-W41C, rHA1-K299R/NA-L35R/M2-W41C, and rWT, respectively. Next, CRFK cells were infected with each recombinant virus, and the virus titers were measured. The plaque sizes were similar in all the eight generated viruses in MDCK cells. The titers of all the four single amino acid mutants were significantly higher than that of rWT virus at most time points. Both the HA mutants, rHA1-K299R and rHA2-T107I, enhanced virus growth at higher levels than that by rNA-L35R and rM2-W41C (Fig 2A). In addition, all the double or triple mutants grew better than single amino acid mutants, indicating synergistic effects of the mutations (Fig 2B). These data suggest that each mutation has an independent mechanism of virus growth in CRFK cells and is involved in the CIV adaptation to CRFK cells.

## Fusion activity of the HA mutants

Amino acid positions, 299 in HA1 and 107 in HA2, were located near and inside the stalk region of the HA. Therefore, we suspected that these mutations would affect membrane fusion activity of the HA. We conducted a cell fusion assay by co-transfecting wild-type or mutant HA expression plasmids and a Venus expression plasmid into MDCK cells. At 24 h post transfection, cells were exposed to several low-pH buffers, and membrane fusion was observed (Fig 3). We did not find any fused cells in wild-type and mutant HA expressing cells at a pH range of more than 5.8. In contrast, at pH 5.0, 5.2, and 5.4, fused cells were observed in both the wild-type and mutant HA expressing cells. At pH 5.6, fused cells were observed only in mutant T107I HA expressing cells. These data suggest that HA2-T107I mutation widens the optimal pH range for virus membrane fusion in MDCK cells.

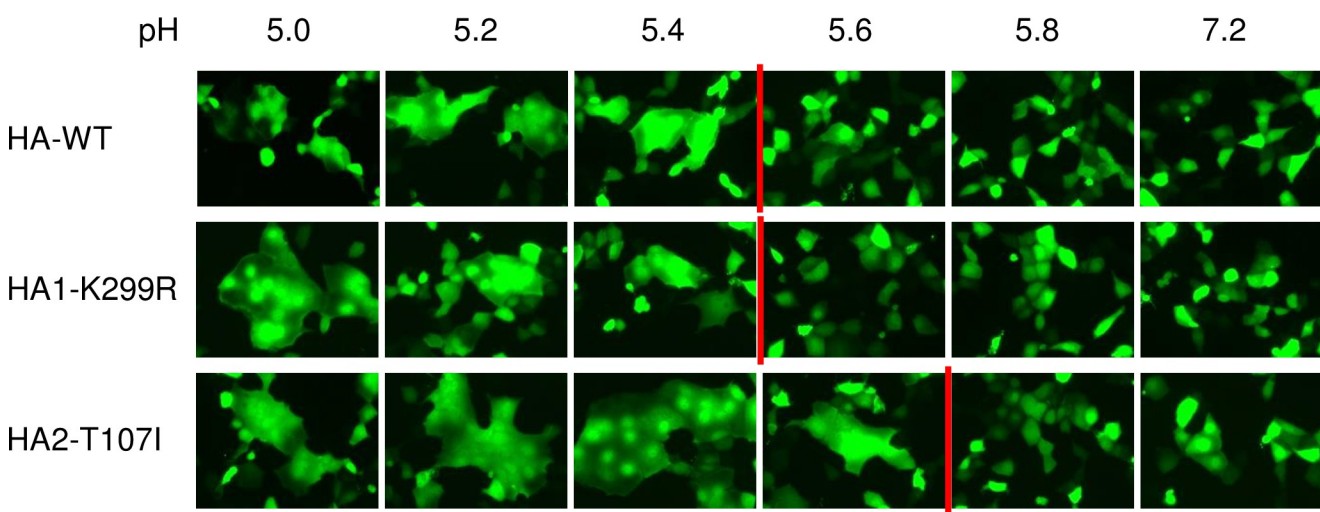

**Fig 3. pH range of syncytium formation of the mutant HAs in MDCK cells.** MDCK cells were co-transfected with a mutant (HA1-K299R or HA2-T107I) or wild-type (HA-WT) HA expression plasmid and a Venus expression plasmid. After treatment with the indicated pH buffers, the cells were observed under a fluorescence microscope. The red lines indicate the boundaries of pH at which cell fusion occurred. The assay was conducted three times independently and the representative data are shown.

### Thermostability of the HA mutants

To investigate a mechanism why HA1-K299R and HA2-T107I mutations increased virus growth in CRFK cells, we analyzed the thermostability of the HA mutant viruses. Each virus was diluted, incubated at 42°C for 0, 15, 30, 60, 90, and 120 min, and the reduction of virus titers was measured. We observed that the reduction rates in the titers were not as marked in the HA mutants, compared to those in rWT virus (Fig 4). These data suggest that both the HA1-K299R and HA2-T107I mutations confer thermostability to the CIV, leading to its enhanced growth in CRFK cells.

### Growth of the mutants in MDCK cells

To examine whether the mutations affect the virus growth in canine cells, we assessed growth kinetics of the mutants in MDCK cells. Interestingly, the viruses possessing HA1-K299R or HA2-T107I mutations grew better than rWT at some time points, albeit small differences were observed in their titers. In contrast, the rNA-L35R and rM-W41C viruses exhibited similar growth properties to rWT (Fig 5A). Growth kinetics with a combination of the mutations confirmed these observations (Fig 5B). These data demonstrated that mutations required for feline cell-adaptation of the CIV have little effect on the virus growth in canine cells. Notably, all of the mutants were unable to replicate in human-derived A549 or monkey-derived Vero cells, similar to rWT (data not shown).

### Growth of the mutants in fcwf-4 cells

We examined growth of the mutants in other type of feline cells – macrophage-like fcwf-4 cells. All the viruses possessing HA1-K299R or HA2-T107I mutations grew better than rWT as did in epithelial-type CRFK cells. In contrast, growth kinetics of the viruses possessing single NA-L35R or M-W41C mutations were similar to those of rWT (Fig 6A). However, NA-L35R mutation enhanced the viral growth in combination with HA1-K299R mutation in a synergistic manner (Fig 6B). These data indicated that HA1-K299R or HA2-T107I mutations were primary determinants for feline cell-adaptation of the CIV.

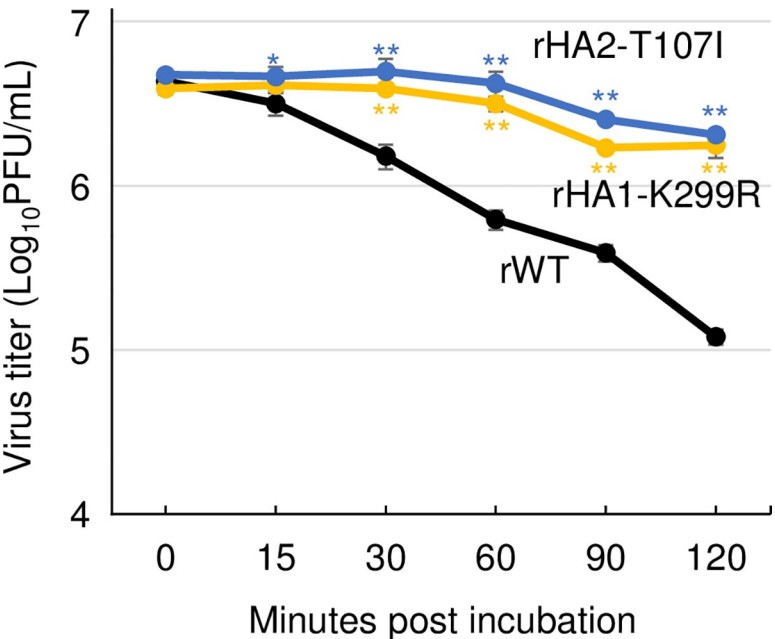

**Fig 4. Thermal stability of the mutant viruses.** The mutant (rHA1-K299R and rHA1-T107I) and wild-type (rWT) viruses were diluted to $4 \times 10^6$ PFU/mL and aliquoted. After the aliquots were heated at 42°C for 0, 15, 30, 60, 90, or 120 min, virus titers were measured. Error bars represent SD of the means from three independent experiments. Statistical differences in the titers between the mutant viruses and wild-type virus (rWT) were assessed by two-tailed Student's t-test ($^*p<0.05$, $^{**}p<0.01$).

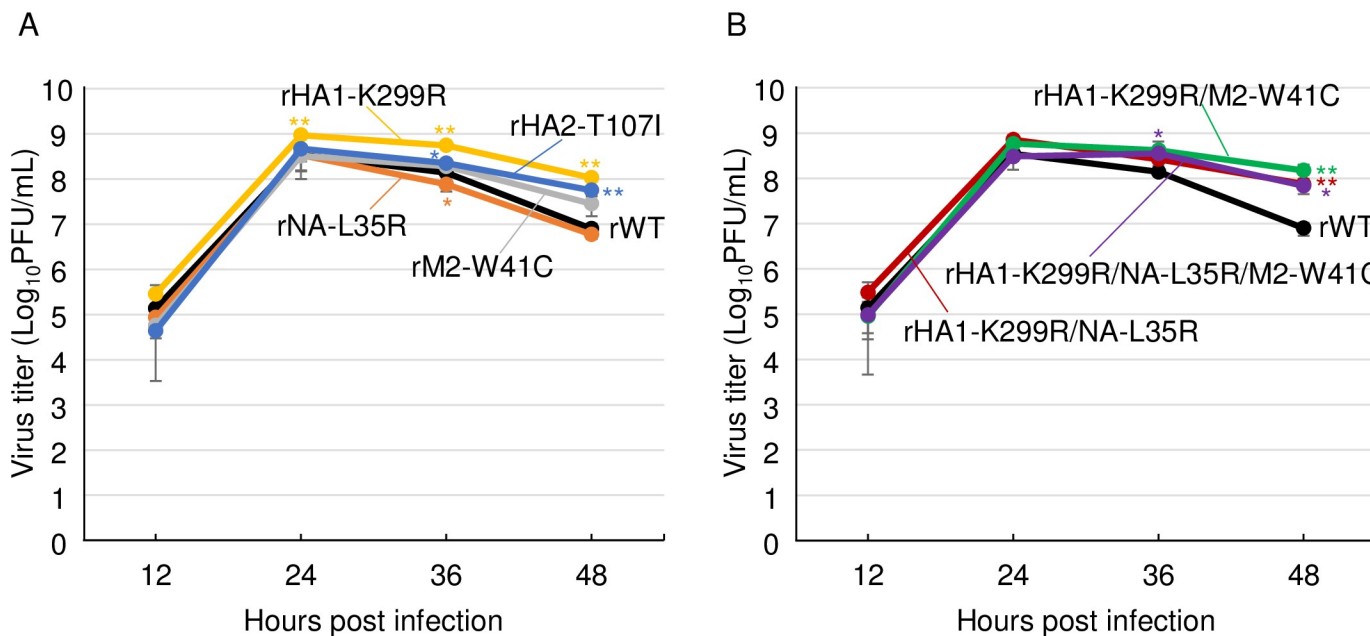

**Fig 5. Growth kinetics of the mutants in MDCK cells.** MDCK cells were infected with recombinant viruses possessing (A) single mutation (rHA1-K299R, rHA2-T107I, rNA-L35R, or rM2-W41C), (B) multiple mutations (rHA1-K299R/NA-L35R, rHA1-K299R/M2-W41C, or rHA1-K299R/NA-L35R/M2-W41C), or recombinant wild-type (rWT) virus at an moi of 0.001. The virus titers were measured at each time point. Error bars represent SD of the means from three independent experiments. Statistical differences in the growth between the mutant viruses and rWT were assessed by two-tailed Student's t-test ($^*p<0.05$, $^{**}p<0.01$).

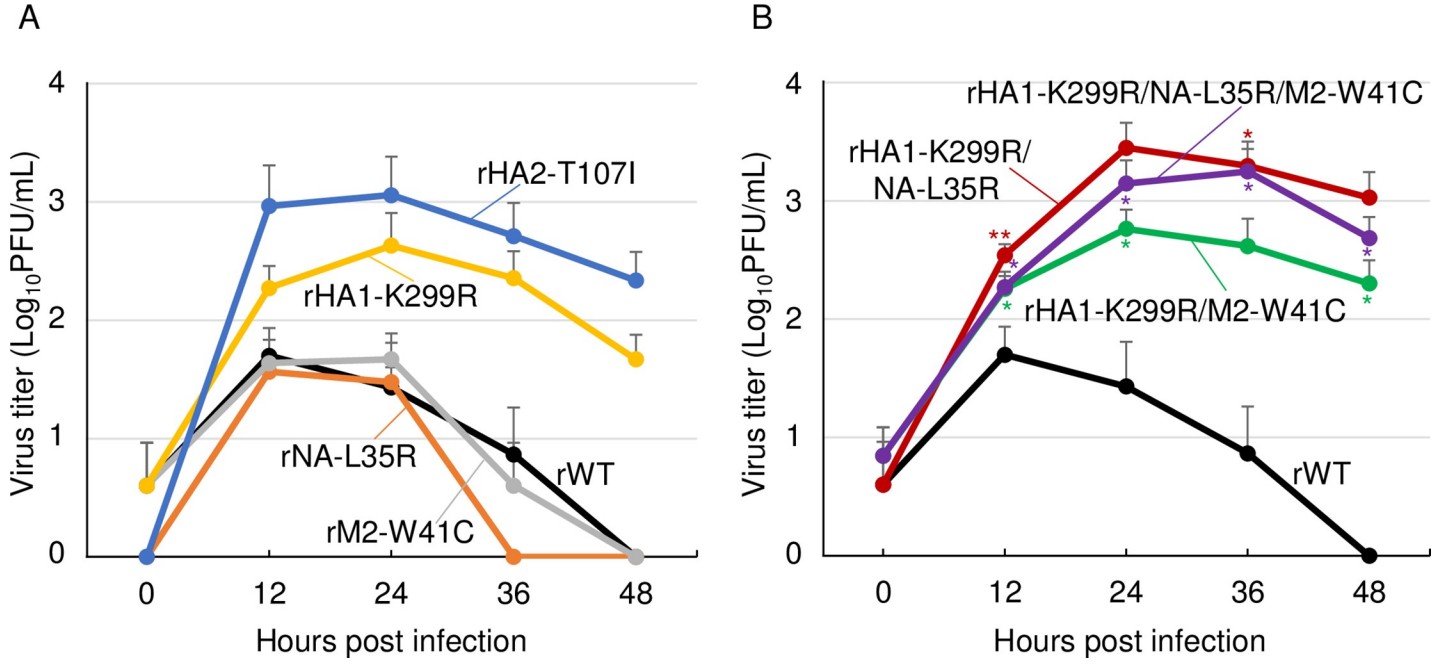

**Fig 6. Growth kinetics of the mutants in fcwf-4 cells.** Fcwf-4 cells were infected with recombinant viruses possessing (A) single mutation (rHA1-K299R, rHA2-T107I, rNA-L35R, or rM2-W41C), (B) multiple mutations (rHA1-K299R/NA-L35R, rHA1-K299R/M2-W41C, or rHA1-K299R/NA-L35R/M2-W41C), or recombinant wild-type (rWT) virus at an moi of 0.01. The virus titers were measured at each time point. Error bars represent SD of the means from three independent experiments: only upward error bars are represented. The statistical differences in the growth between the mutant viruses and rWT were assessed by two-tailed Student's t-test ($^*$p<0.05, $^{**}$p<0.01).

## Discussion

The recent spread of CIVs in some Asian countries and the USA raises a public health concern that the virus could be transmitted to humans directly or through intermediate hosts, causing a new pandemic [20]. Cats are probable candidates that could act as intermediate hosts, since they have a close relationship to humans as companion animals. Previous reports showing H3N2 CIV transmission to cats may accentuate this idea [28,29]. If a CIV adapts to cats and spreads in the cat population, the viruses have more opportunities to be transmitted to humans, as they can infect not only cats but also dogs. However, the molecular basis of H3N2 CIV adaptation to cats has not been elucidated. Here, we generated feline cell-adapted CIVs and revealed that both HA1-K299R and HA2-T107I mutations enhanced virus growth in feline cell lines, by altering the optimal pH for viral membrane fusion and/or promoting thermostability of the virus. Previous studies have shown that such alterations of HAs may play a role in the adaptation of H1N1, H5N2, and H9N2 viruses to different host cells and animals [35,37–39]. In addition, we found that NA-L35R and M-W41C mutations could improve CIV growth in feline cell lines, albeit less efficiently, compared to HA mutations. These findings suggest that HA1-K299R, HA2-T107I, NA-L35R, and M-W41C mutations might be responsible for CIV adaptation to cats, although studies need to be conducted by experimentally infecting the feline cell-adapted CIVs into cats.

Two H3N2 CIVs were isolated from cats at animal shelters in Korea in 2010, revealing direct transmission of the virus from dogs to cats [28,29]. In one study, CIVs were transmitted from dogs in the same shelter, where the virus was circulating among the dogs [28]. In this paper, 18 amino acid differences were found in genomes between the dog and cat isolates, all of which were not observed in our study. Among them, two HA mutations, HA1-S107P and

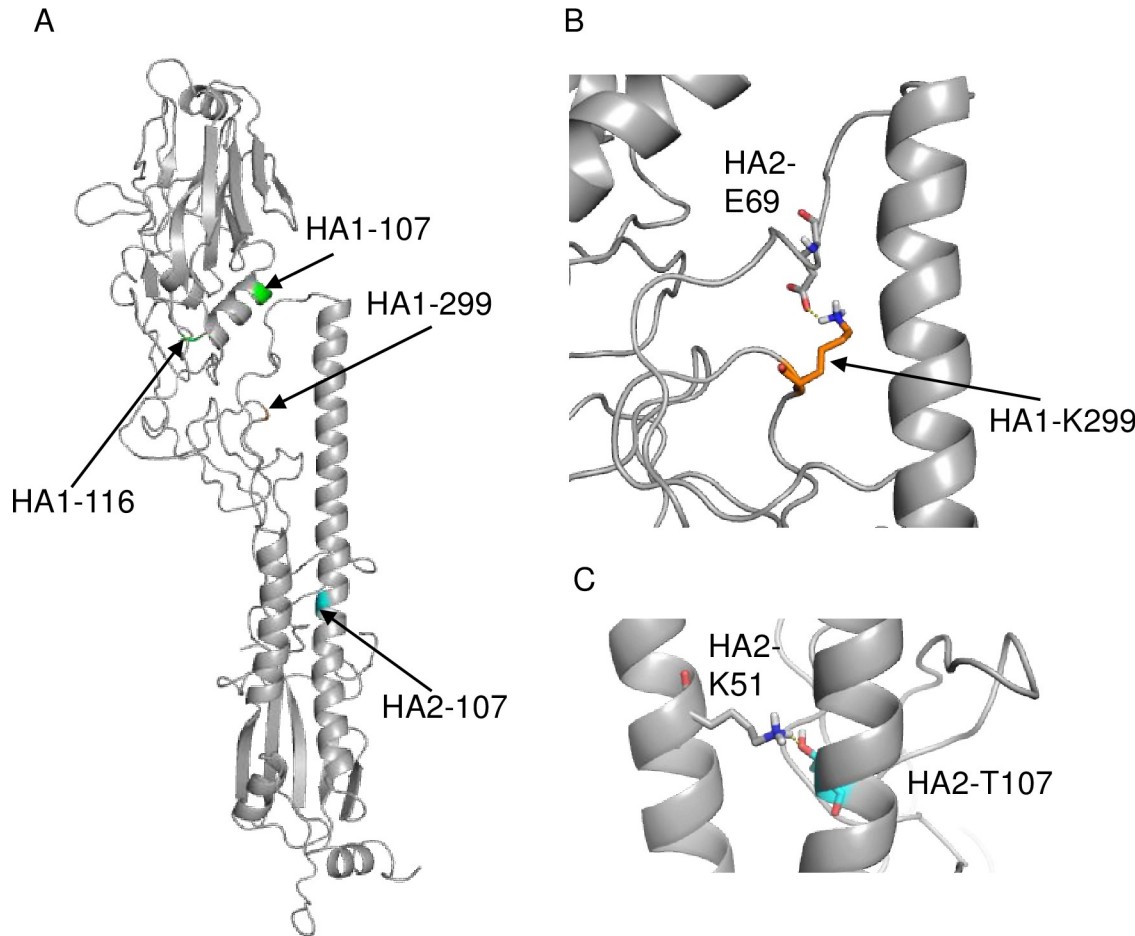

**Fig 7. 3D structure of H3-HA protein.** The 3D structure of HA of A/Aichi/2/1968 (H3N2) (PDB: 1HGG) is shown. (A) Amino acid positions, HA1-299 and HA2-107, at which mutations were observed in the CRFK-adapted CIVs, are colored orange or cyan, respectively. Amino acid positions, 107 and 116, in HA1, at which substitutions were observed in the feline isolates of CIV [28], are colored green. (B) Close-up view of the area surrounding HA1-299 is shown. A polar contact is formed between HA1-K299 and HA2-E69. (C) Close-up view of the area surrounding HA2-107 is shown. A polar contact is formed between HA2-T107 and HA2-K51.

-S116G (H3 numbering), were found in cat isolates, implicating that these mutations might be responsible for CIV adaptation to cats. The amino acid positions at these mutations were located at bottom of head region of the HA. Interestingly, these locations are near the position 299 in HA1-K299R mutation found in our study (Fig 7A). An undefined mechanism related to these HA1 mutations may be associated with CIV adaptation to cats.

Although a previous study showed that K299G or K299E mutations in HA1 of X-31 strain (H3 subtype) increased the optimal pH for conformational change into membrane fusion [40], CIV HA1-K299R mutation observed in our study did not affect the optimal pH of the membrane fusion (Fig 3). Instead, this mutation lead to thermal resistance of the mutant virus (Fig 4). Lysine at position 299 of H3 HA forms a polar contact with HA2-69E [41] (Fig 7B), and this contact strength may have increased by lysine to arginine substitution, because arginine is more negatively charged than lysine. Such a change in polar contact strength could have increased the stability of the mutant virus. On the contrary, HA2-T107I mutation increased both the optimal pH of membrane fusion as well as thermal resistance of virus. Previous studies showed that polar contacts, which are formed between amino acids around HA2 position

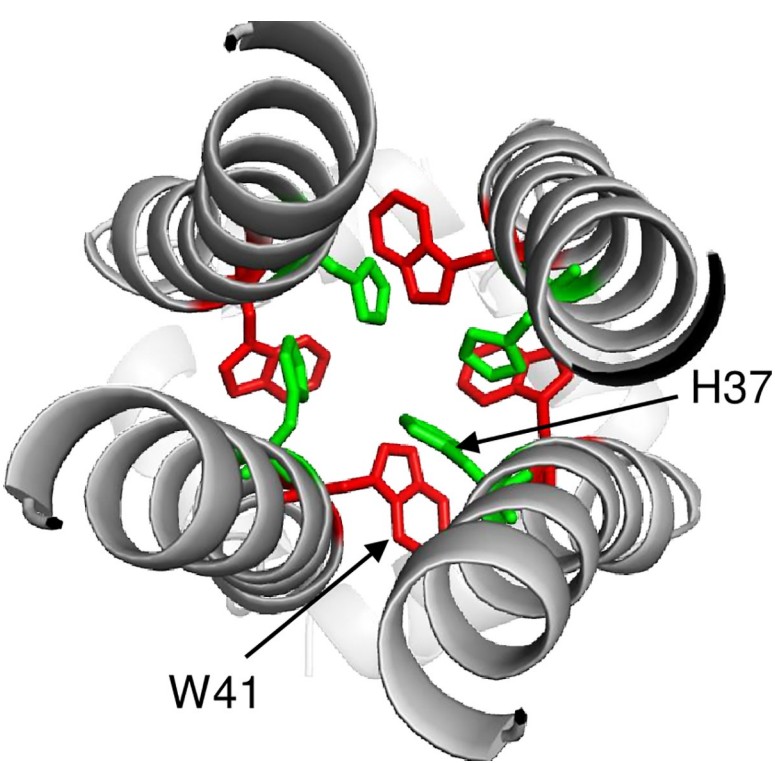

**Fig 8. 3D structure of M2 protein.** The 3D structure of M2 of A/Udorn/1972 (H3N2) (PDB: 2RLF) is shown. H37 and W41, which possess cation-$\pi$ interaction, are colored green and red, respectively.

107 and HA2 positions 39–55 in α-helix structure, markedly affected the optimal pH for membrane fusion [42–44]. In our study, HA2-T107I mutation may have lost a polar contact with HA2-51K (Fig 7C), thereby increasing the optimal pH for membrane fusion. Additional analyses are required to elucidate the mechanisms of changes in thermal stability by mutations.

Amino acids at position 31–35 of NA transmembrane domain are involved in its raft association [45]. Therefore, NA-L35R mutation found in the mutants may affect the raft association of NA, resulting in a change in the amount of this protein in the virions. The change in NA content in the virions may have altered the HA-NA functional balance, leading to high growth of the mutant in feline cells.

An ion channel M2 protein located on virus envelope is activated under acidic pH condition of endosomes [46]. A histidine residue at position 37 and a tryptophan residue at position 41 have cation-$\pi$ interaction (Fig 8) [47]. This interaction is important for the proton-selective ion channel activity of M2, and M2-41W residue, which plays a role as a gate-keeper to prevent spilling out of protons from inside the acidic virions [48,49]. M2-W41C mutation caused high membrane conductance under high pH condition of an external virion [49], suggesting that the M2-W41C mutant could allow protons to flow into the virion even at a higher pH of the endosomes. Together with HA2-T107I mutation that raises the pH condition for membrane fusion, CIVs may have been adapted to endosomal pH environment in the feline cells by HA as well as M2 mutations.

In conclusion, we identified multiple mutations required for adaptation of H3N2 CIV to feline cells. Although a study with experimental infection of feline cell-adapted viruses in cats is required, the mutations found in these viruses may contribute to understand the mechanism on cross-species transmission of CIVs and risk assessment for pandemic preparedness.

## Acknowledgments

We thank Dr. Kathy Toohey-Kurth (Veterinary Diagnostic Laboratory, School of Veterinary Medicine, University of Wisconsin, USA) for A/canine/Madison/5/2015 (H3N2) virus.

## Author Contributions

**Conceptualization:** Haruhiko Kamiki, Shin Murakami, Taisuke Horimoto.

**Data curation:** Haruhiko Kamiki, Shin Murakami.

**Formal analysis:** Haruhiko Kamiki, Shin Murakami.

**Investigation:** Haruhiko Kamiki, Hiromichi Matsugo, Hiroho Ishida, Tomoya Kobayashi-Kitamura, Wataru Sekine, Akiko Takenaka-Uema, Shin Murakami, Taisuke Horimoto.

**Project administration:** Taisuke Horimoto.

**Supervision:** Taisuke Horimoto.

**Writing – original draft:** Haruhiko Kamiki.

**Writing – review & editing:** Shin Murakami, Taisuke Horimoto.

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
