## [Decision Letter · Decision Letter 0]

2 Aug 2019

PONE-D-19-18114

Adaptation of H3N2 canine influenza virus to feline cell culture

PLOS ONE

Dear Prof. Horimoto,

Thank you for submitting your manuscript to PLOS ONE. After careful consideration, we feel that it has merit but does not fully meet PLOS ONE’s publication criteria as it currently stands. Therefore, we invite you to submit a revised version of the manuscript that addresses the points raised during the review process.

We would appreciate receiving your revised manuscript by Sep 16 2019 11:59PM. To enhance the reproducibility of your results, we recommend that if applicable you deposit your laboratory protocols in protocols.io, where a protocol can be assigned its own identifier (DOI) such that it can be cited independently in the future. For instructions see: http://journals.plos.org/plosone/s/submission-guidelines#loc-laboratory-protocols

We look forward to receiving your revised manuscript.

Kind regards,

Kensuke Hirasawa, PhD

Academic Editor

PLOS ONE

Journal Requirements:

Reviewers' comments:

Reviewer's Responses to Questions

**Comments to the Author**

1. Is the manuscript technically sound, and do the data support the conclusions?

Reviewer #1: Partly

Reviewer #2: Yes

2. Has the statistical analysis been performed appropriately and rigorously? 

Reviewer #1: Yes

Reviewer #2: Yes

3. Have the authors made all data underlying the findings in their manuscript fully available?

Reviewer #1: No

Reviewer #2: Yes

4. Is the manuscript presented in an intelligible fashion and written in standard English?

Reviewer #1: Yes

Reviewer #2: Yes

5. Review Comments to the Author

Reviewer #1: This manuscript describes adaptation of an influenza A virus strain that was isolated from a dog to a cat derived cell line. The authors analyzed genetic changes and their effects on the virus phenotypes. They initially identified the mutations after a serial passage, then introduced the found mutations individually or in combination into the genetic background of the parental dog virus genome by reverse genetics. The phenotypes of these recombinant viruses were tested in vitro. The study is original, methodology was sound and the results were properly analyzed and presented. However, I would ask the authors several questions regarding the motivation and the design of this study, and the interpretation of the results:

1. The authors tried to identify the required mutation for a dog influenza A virus to adapt to cat. Why the authors thought cat-adapted influenza A virus would be more dangerous than dog virus to human (line 73-74 & 311-312)? The authors stated that “cats are probable candidates that could act as intermediate hosts, …” (line 308-309). Aren’t dogs or other mammalian species, as well?

2. The authors passed dog derived influenza A virus in CRFK cells to adapt the virus to cat cells. Are these viruses really better fit to cat, though they stated “… CIV adaptation to cats.” (line 319)? Good virus replication in a host is essential for the virus spread to another host, however, is there any evidence that supports parallel relationship between the replication efficiency and the transmission efficiency?

3. The authors presented evidence that HA1-K299R and HA2-T107I mutations enhance virus growth in cat derived cells, alter the optimal (threshold?) pH for membrane fusion and thermostability. However, these do not necessarily mean the latter alterations were the cause for the former enhanced virus growth in cat cells. Why the altered optimal pH for fusion and thermostability are preferable for the virus to grow in cat (but not dog) cells (line 315)? Isn’t higher thermostability preferable for the virus replication in dog cells in vitro, as well? Is there any evidence or literature that support the authors’ speculation?

4. The authors compared a cat isolate in a literature to their own CRFK adapted virus (P. 15, last paragraph). When an influenza A virus adapted to another host species NS1 and PB2 were mutated in previous literatures (ref. 26 and J. Virol. 84, 10606-). Is there any possible reason why neither gene did not mutate in your experiment?

5. The authors stated “CIVs may have been adapted to endosomal pH environment in feline cells …” (line 369). How the endosomal environment is different between in CRFK and in MDCK? Do you have any information? This would also help answer the question 3 above.

Minor points

• Line 51-64: This paragraph should be better organized. Transmissions cannot be isolated (line 59-60).

• Line 234: Figures should appear in the text in numerical order; Fig. 7A should not appear before Fig. 3.

• Fig. 3: The pictures need better focus.

Reviewer #2: This manuscript described that canine influenza virus H3N2 was atapted to feline cells and the mechanism of adaptation was analyzed. The authors discussed that two mutations are responseible for adaptation of H3N2 canine influenza virus in cats and the mechanisms were thermal resistance and pH requirement. However, my major comments should be addressed.

Major comments

These mutations might occur after adaptation to in vitro cell culture, but not only to feline cell culture. It should be denied that these mutations occurred after adaptation to cell culture. Canine influenza virus H3N2 should be adapted to the other mammalian cell lines.

These recombinant viruses should be analyzed in the other cells line, but not only in MDCK, CRFK and fewf-4 cells.

6. PLOS authors have the option to publish the peer review history of their article (what does this mean?). If published, this will include your full peer review and any attached files.

Reviewer #1: No

Reviewer #2: No

---

## [Author Response · Author response to Decision Letter 0]

1 Sep 2019

Responses to Reviewer #1: 

“1. The authors tried to identify the required mutation for a dog influenza A virus to adapt to cat. Why the authors thought cat-adapted influenza A virus would be more dangerous than dog virus to human (line 73-74 & 311-312)? The authors stated that “cats are probable candidates that could act as intermediate hosts, …” (line 308-309). Aren’t dogs or other mammalian species, as well?”

We apologize for the unclear statement. We intended to argue that both companion animals, dogs and cats, are potential candidates that can act as intermediate hosts, due to their close contact to humans. To clarify this point, we altered the sentence in the revised manuscript (lines 75/319-320) as follows:

“If a CIV ……, the viruses …., as they infect not only cats but also dogs.’’

“2. The authors passed dog derived influenza A virus in CRFK cells to adapt the virus to cat cells. Are these viruses really better fit to cat, though they stated “… CIV adaptation to cats.” (line 319)? Good virus replication in a host is essential for the virus spread to another host, however, is there any evidence that supports parallel relationship between the replication efficiency and the transmission efficiency?”

We agree with this comment. However, to make this point clear, experimental infection and transmission studies with feline cell-adapted viruses in cats are required. We are planning to do these experiments in cats as a next step. Therefore, we rephrased the sentence in the revised manuscript (line 329-330) as follows: 

“These findings……mutations may be responsible for CIV adaptation to cats, although studies need to be conducted by experimentally infecting the feline cell-adapted CIVs into cats.”.

“3. The authors presented evidence that HA1-K299R and HA2-T107I mutations enhance virus growth in cat derived cells, alter the optimal (threshold?) pH for membrane fusion and thermostability. However, these do not necessarily mean the latter alterations were the cause for the former enhanced virus growth in cat cells. Why the altered optimal pH for fusion and thermostability are preferable for the virus to grow in cat (but not dog) cells (line 315)? Isn’t higher thermostability preferable for the virus replication in dog cells in vitro, as well? Is there any evidence or literature that support the authors’ speculation?”

We agree with this comment. However, we have no direct evidence to support our speculation. Therefore, we added a sentence with references reporting that the changes in optimal HA for membrane fusion or thermostability could be related to replication and transmission of the viruses to other host cells and animals in the revised manuscript (lines 324-325) as follows: 

“Previous studies have shown that such alterations of HAs may play a role in the adaptation of H1N1, H5N2, and H9N2 viruses to different host cells and animals [35,37–39].”

We referred to these papers in the revised manuscript.

37. Sang et al., Adaptation of H9N2 AIV in guinea pigs enables efficient transmission by direct contact and inefficient transmission by respiratory droplets. Sci Rep. 2015;5: 15928.

38. Klein et al., Stability of the influenza virus hemagglutinin protein correlates with evolutionary dynamics. mSphere. 2018;3: e00554-17.

39. Singanayagam et al., Influenza virus with increased pH of hemagglutinin activation has improved replication in cell culture but at the cost of infectivity in human airway epithelium. J Virol. 2019;93: e00058-19.

“4. The authors compared a cat isolate in a literature to their own CRFK adapted virus (P. 15, last paragraph). When an influenza A virus adapted to another host species NS1 and PB2 were mutated in previous literatures (ref. 26 and J. Virol. 84, 10606-). Is there any possible reason why neither gene did not mutate in your experiment?”

As the reviewer has rightly pointed out, several mutations in the internal genes such as IFN-antagonized NS1 and polymerase subunit PB2 have been frequently observed, especially during in vivo adaptation of avian viruses to mammalian hosts. Contrastingly, our mutants were obtained through in vitro cell culture passages of a mammalian (dog) virus in mammalian (cat) cells. Therefore, we speculate that the difference in the methods adopted for viral adaptation between the previous studies and our study could be a possible reason why these internal genes did not mutate in our mutants. We have added a sentence addressing this point in the revised manuscript (lines 200-202).

“5. The authors stated “CIVs may have been adapted to endosomal pH environment in feline cells …” (line 369). How the endosomal environment is different between in CRFK and in MDCK? Do you have any information? This would also help answer the question 3 above.”

We could not obtain any information on the difference in the endosomal pH environments between CRFK and MDCK cells. Therefore, we just made a mention of this issue in the manuscript (lines 380-381) as follows: 

“CIVs may have been adapted to endosomal pH environment in feline cells by HA as well as M2 mutations.” 

Minor points

“Line 51-64: This paragraph should be better organized. Transmissions cannot be isolated (line 59-60).”

We rearranged this paragraph in the revised manuscript (lines 51-65) as follows: 

“Reports have shown that several human and avian influenza viruses can infect domestic cats. Serological surveys suggested that H3N2 human seasonal viruses have been transmitted to cats [21,22] and H1N1pdm2009 human virus managed to infect a cat colony, causing severe clinical diseases with fatalities [23]. The highly pathogenic H5N1 avian influenza virus (HPAIV) naturally infects domestic cats and causes severe illnesses [24]. In 2016, a low pathogenic strain of H7N2 AIV was transmitted to cats in an animal shelter and a veterinarian, who took care of these cats, got infected with the virus transmitted from the infected cats, causing mild respiratory symptoms [25]. In addition, under experimental conditions, domestic cats have been known to be susceptible to other influenza viruses such as H2N2 human, H7N7 seal, and H7N3 avian viruses [3,26,27]. 

 Recently, transmission of H3N2 CIV into cats was also reported in two animal shelters in South Korea, where dogs and cats were accommodated together, resulting in 21.7% and 40% mortality of the cats in each shelter, respectively; the affected cats exhibited severe respiratory signs [28, 29]. The transmission of H3N2 CIV into cats was also reported in China [20].”

“Line 234: Figures should appear in the text in numerical order; Fig. 7A should not appear before Fig. 3.”

We deleted “Fig. 7A” in the revised manuscript (line 239), as it was not necessary in the paper.

 “Fig. 3: The pictures need better focus.”

We apologize for this. However, we were unable to replace the pictures as we have used the best images we had in Fig. 3. 

 

Responses to Reviewer #2: 

“These mutations might occur after adaptation to in vitro cell culture, but not only to feline cell culture. It should be denied that these mutations occurred after adaptation to cell culture. Canine influenza virus H3N2 should be adapted to the other mammalian cell lines.”

We were unable to obtain any CIVs adapted to human-derived A549 or monkey-derived Vero cells. In fact, even wild-type CIVs hardly replicated in these cells. We have added this observation in the revised manuscript (lines 186-187).

“These recombinant viruses should be analyzed in the other cells line, but not only in MDCK, CRFK and fewf-4 cells.”

Feline CRFK cell-adapted CIVs were unable to replicate in human A549 and monkey Vero cells as was the parent virus. We have added this observation in the revised manuscript (lines 281-283).

---

## [Decision Letter · Decision Letter 1]

24 Sep 2019

Adaptation of H3N2 canine influenza virus to feline cell culture

PONE-D-19-18114R1

Dear Dr. Horimoto,

We are pleased to inform you that your manuscript has been judged scientifically suitable for publication and will be formally accepted for publication once it complies with all outstanding technical requirements.

With kind regards,

Kensuke Hirasawa, PhD

Academic Editor

PLOS ONE

Additional Editor Comments (optional):

Reviewers' comments:

Reviewer's Responses to Questions

**Comments to the Author**

1. If the authors have adequately addressed your comments raised in a previous round of review and you feel that this manuscript is now acceptable for publication, you may indicate that here to bypass the “Comments to the Author” section, enter your conflict of interest statement in the “Confidential to Editor” section, and submit your "Accept" recommendation.

Reviewer #1: All comments have been addressed

Reviewer #2: All comments have been addressed

2. Is the manuscript technically sound, and do the data support the conclusions?

Reviewer #1: Yes

Reviewer #2: Yes

3. Has the statistical analysis been performed appropriately and rigorously? 

Reviewer #1: Yes

Reviewer #2: Yes

4. Have the authors made all data underlying the findings in their manuscript fully available?

Reviewer #1: Yes

Reviewer #2: Yes

5. Is the manuscript presented in an intelligible fashion and written in standard English?

Reviewer #1: Yes

Reviewer #2: Yes

6. Review Comments to the Author

Reviewer #1: (No Response)

Reviewer #2: This revised manuscript was modified according to our comments. Now I agree this manuscript is acceptable for publication.

7. PLOS authors have the option to publish the peer review history of their article (what does this mean?). If published, this will include your full peer review and any attached files.

Reviewer #1: No

Reviewer #2: No

---

## [Editor Report · Acceptance letter]

2 Oct 2019

PONE-D-19-18114R1 

Adaptation of H3N2 canine influenza virus to feline cell culture 

Dear Dr. Horimoto:

I am pleased to inform you that your manuscript has been deemed suitable for publication in PLOS ONE. Congratulations! Your manuscript is now with our production department. 

With kind regards,

on behalf of

Dr. Kensuke Hirasawa 

Academic Editor

PLOS ONE